

# Implementation of artificial intelligence for the detection of cutaneous melanoma within a primary care setting: prevalence and types of skin cancer in outdoor enthusiasts

Ian J. Miller[1,2], Michael Stapelberg[1,2,3], Nedeljka Rosic[1,2], Jeremy Hudson[1,2,4], Paul Coxon[4], James Furness[5], Joe Walsh[6,7] and Mike Climstein[1,2,5,8]

[1] Aquatic Based Research, Southern Cross University, Bilinga, Queensland, Australia
[2] Faculty of Health, Southern Cross University, Bilinga, Queensland, Australia
[3] Specialist Suite, John Flynn Hospital, Tugun, Queensland, Australia
[4] North Queensland Skin Centre, Townsville, Queensland, Australia
[5] Water Based Research Unit, Bond University, Robina, Queensland, Australia
[6] Sport Science Institute, Sydney, NSW, Australia
[7] AI Consulting Group, Sydney, NSW, Australia
[8] Physical Activity, Lifestyle, Ageing and Wellbeing Faculty Research Group, University of Sydney, Sydney, NSW, Australia

Corresponding authors
Ian J. Miller,
i.miller.11@student.scu.edu.au
Mike Climstein,
michael.climstein@scu.edu.au

## ABSTRACT

**Background:** There is enthusiasm for implementing artificial intelligence (AI) to assist clinicians detect skin cancer. Performance metrics of AI from dermoscopic images have been promising, with studies documenting sensitivity and specificity values equal to or superior to specialists for the detection of malignant melanomas (MM). Early detection rates would particularly benefit Australia, which has the worlds highest incidence of MM *per capita*. The detection of skin cancer may be delayed due to late screening or the inherent difficulty in diagnosing early skin cancers which often have a paucity of clinical features and may blend into sun damaged skin. Individuals who participate in outdoor sports and recreation experience high levels of intermittent ultraviolet radiation (UVR), which is associated with the development of skin cancer, including MM. This research aimed to assess the prevalence of skin cancer in individuals who regularly participate in activities outdoors and to report the performance parameters of a commercially available AI-powered software to assess the predictive risk of MM development.

**Methods:** Cross-sectional study design incorporating a survey, total body skin cancer screening and AI-embedded software capable of predictive scoring of queried MM.

**Results:** A total of 423 participants consisting of surfers ($n = 108$), swimmers ($n = 60$) and walkers/runners ($n = 255$) participated. Point prevalence for MM was highest for surfers (6.48%), followed by walkers/runners (4.3%) and swimmers (3.33%) respectively. When compared to the general Australian population, surfers had the highest odds ratio (OR) for MM (OR 119.8), followed by walkers/runners (OR 79.74), and swimmers (OR 61.61) rounded out the populations. Surfers and swimmers reported comparatively lower lifetime hours of sun exposure (5,594 and 5,686, respectively) but more significant amounts of activity within peak ultraviolet

index compared with walkers/runners (9,554 h). A total of 48 suspicious pigmented lesions made up of histopathology-confirmed MM ($n$ = 15) and benign lesions ($n$ = 33) were identified. The performance of the AI from this clinical population was found to have a sensitivity of 53.33%, specificity of 54.44% and accuracy of 54.17%. **Conclusions:** Rates of both keratinocyte carcinomas and MM were notably higher in aquatic and land-based enthusiasts compared to the general Australian population. These findings further highlight the clinical importance of sun-safe protection measures and regular skin screening in individuals who spend significant time outdoors. The use of AI in the early identification of MM is promising. However, the lower-than-expected performance metrics of the AI software used in this study indicated reservations should be held before recommending this particular version of this AI software as a reliable adjunct for clinicians in skin imaging diagnostics in patients with potentially sun damaged skin.

## INTRODUCTION

Popular outdoor past-time activities in Australia include surfing, swimming and walking, with over 10 million people estimated to participate in these activities (*SportAus Ausplay, 2019*). For example, recreational walking is Australia's most popular outdoor activity, with an estimated 8.7 million Australians participating (*SportAus Ausplay, 2019*). Swimming is Australia's most popular aquatic activity by numbers, with an estimated six million individuals recognized as actively participating (*SportAus Ausplay, 2019*). With regard to surfing, it has been estimated that there are over 2.7 million surfers in Australia (*Stark, 2013*) and The Australian Sports Commission estimates there are 196,000 Australians, 15 years of age and older, that have taken up surfing since 2020 (*McCubbing, 2022*). Individuals who participate in these outdoor activities, aquatic or land-based, are exposed to substantially higher ultraviolet radiation (UVR) exposure (*Snyder et al., 2020*) than those individuals completing indoor recreational activities.

Exposure to UVR is well recognized as a mechanism for the development of precancerous skin lesions such as actinic keratosis (AK) and cancerous lesions including malignant melanomas (MM) and keratinocyte carcinomas (KC), the most prevalent being basal cell carcinomas (BCCs) and squamous cell carcinomas (SCCs) (*Olsen et al., 2015*). The UV index individuals within Australia experience is greater than those individuals residing within the United States and Europe, based on the continent's geographical locations (*Umar & Tasduq, 2022*). Furthermore, the protective effect the ozone layer provides has been well documented to be thinner above Australia, resulting in greater levels of UVR reaching the 'Earth's surface in this location (*Umar & Tasduq, 2022*). Therefore, individuals within Australia are likely to experience more intense doses of UVR over a shorter amount of time than many fair-skinned nations.

Although the prevalence of skin cancer in outdoor walkers within Australia is yet to be established, skin cancer point-prevalence in surfers and swimmers has previously been reported as 76 times and 19 times, respectively, to that of the general population (*Climstein et al., 2022*). To limit exposure to UVR, individuals who perform outdoor sports and recreational activities are advised to adhere to sun protection strategies, including the regular application of sunscreen and wearing protective clothing (*Snyder et al., 2020*). Given the increased UVR exposure in outdoor enthusiasts, both aquatic and land-based, it is recommended that primary care physicians regularly screen these individuals (*Climstein et al., 2016*).

The incidence of skin cancers detected in Australia has increased over the past 40 years (*Wada et al., 2020*). For MM, the Australian Institute of Health and Welfare (AIHW) reported 27 cases per 100,000 in 1982 and 49 per 100,000 in 2016, which represents an 81% increase in the standardized rate in the Australian general population (*Australian Institute of Health and Welfare, 2016*). As of 2022, the most recent standardized rate from AIHW is 54.1 cases per 100,000 individuals (*Australian Institute of Health and Welfare, 2022*), a further 10% increase. The increased incidence of skin cancer places further financial burden on the Australian medical system, recently estimated at $426.2 and $397.9 million for KC and MM, respectively (*Gordon et al., 2022*). Factors such as increasing age within Australia and an individual's prior history of KC are strong predictors of increased risk of BCC or SCC in the Australian population (*Pandeya, Olsen & Whiteman, 2017*). Early diagnosis of skin cancer has been shown to result in improved patient outcomes and reduced mortality (*Jones et al., 2022*). The use of AI as a non-invasive screening and diagnostic tool for the early detection of skin cancers is likely to provide the most promise within a primary care setting where the majority of melanomas are diagnosed (*Giavina-Bianchi et al., 2021*).

Machine learning based on convolutional neural networks (CNN) is commonly used as either serial total body photography (TBP) or individual lesion analysis. The promising AI analysis of individual lesions have demonstrated a high sensitivity in diagnosing melanomas from benign lesions (*Jones et al., 2022*). Though a recognized limitation of training an AI algorithm is the program's heavy reliance on a large database of reference images (*Wada et al., 2020*). Current research demonstrates that AI sensitivity is similar to that of trained specialists (*Giavina-Bianchi et al., 2021*; *Winkler et al., 2020*; *Phillips et al., 2019*). *Maron et al. (2020)* investigated the influence of AI on the accuracy of dermatologist's identification of MM from a sample size of 600 melanomas and 600 benign naevi. They reported that with AI support, the mean sensitivity and specificity of the dermatologists increased significantly (60% to 75%, 65% to 73%, respectively).

*Phillips et al. (2019)* investigated over 1,500 images of suspicious skin lesions, which were analyzed by AI and specialists. The AI was found to have a receiver operator characteristic, a plot of the diagnostic accuracy, of 91.8% and 100% sensitivity, whereas the specialists had a lower receiver operator characteristic of 77.8%. The authors concluded that AI support improved the diagnostic accuracy of dermatologists. However, few studies have reported real-world data, rather opting to utilize image datasets as opposed to real-world usage (*Jones et al., 2022*). *Jones et al. (2022)* reported that of all papers in their
electronic search ($n$ = 14,224 studies, 2000 to 2021, inclusive), only two studies used data from a clinical setting, and both studies had a low prevalence of skin cancers.

*MacLellan et al. (2021)* investigated the sensitivity and specificity of teledermatology and noninvasive imaging techniques. The authors found the Fotofinder Moleanalyzer to have superior sensitivity (88.1%) and specificity (78.8%) as compared to other similar imaging technologies (Melafind, 82.5%, 52.4%; Verisente Aura, 21.4%, 86.2%, respectively). In addition, the authors concluded that the Fotofinder had a sensitivity and specificity similar to that of the dermatologist (sensitivity 96.65%, specificity 32.2%) and the device could be used to complement clinical decision-making with regard to identifying MM. However, *MacLellan et al. (2021)* did not report the accuracy of any of the imaging devices. Therefore, there is a gap in the current understanding of the best practice for the implementation of AI in skin cancer screening.

The point-prevalence of skin cancer within surfing and swimming populations was previously done by whole-body screening in a primary care setting (*Climstein et al., 2022*). Published findings of outdoor enthusiasts such as walkers and runners indicate UVR exposure levels that may place them at risk of developing KC and MM (*Duarte et al., 2019*). Prevalence in walkers/runners has been reported by a single study (*Ambros-Rudolph et al., 2006*) in the literature, *Ambros-Rudolph et al. (2006)* investigated 210 Australian marathon runners (166 males, 44 females, mean age 37 years) and they reported no skin lesions suggestive of MM. Eleven percent of the runners were referred to a specialist with lesions suggestive of KC. The prevalence of skin cancer in walkers/runners has not been researched within Australia. The aim of this study was to investigate the point prevalence of AK, KC and MM in surfers, swimmers, walkers/runners by whole-body screening by a skin cancer doctor. The secondary aim of this study was to investigate the accuracy, sensitivity and specificity of a commercial, high-definition dermatoscope that incorporated AI in a primary care setting to assess the predictive risk of MM.

## MATERIALS AND METHODS

### Study design

Southern Cross University's Human Research Ethics committee granted approval for this study (11 May, 2020/47). This cross-sectional study involved a survey followed by whole-body skin cancer screening. Promotion of the study included media (including radio, television, and online newspaper) in conjunction with the in-house promotion within the skin cancer clinics in Southeast Queensland. Additionally, prior to their scheduled skin examination, participants were enlisted on attendance at a skin cancer clinic in Southeast Queensland, GPS coordinates 28°29′07″S and 153°29′17″E.

### Survey

The survey followed the protocol previously described by *Climstein et al. (2022)*. In brief, the survey consisted of four sections: physiological demographics (participant background), activity-specific sun exposure (main type of activity outdoors, total time spent outdoors, time spent during peak UV index), skin cancer prevention strategies (hat or shirt use, sunscreen or zinc use), skin cancer risk and history (number of sunburns in

prior 12 month period, skin complexion, family history of skin cancer, prior history of skin cancer). All surveys were completed in the clinic waiting room prior to undergoing whole-body screening for skin cancer. Any questions arising from the participants pertaining to the survey were answered by either the specialist or the researcher on-site.

## Screening strategy

A commercial, high-resolution digital dermatoscope mole mapping system with in-built artificial intelligence *via* deep learning convolutional neural networks (Moleanalyzer-Pro; FotoFinder Systems GmbH, Bad Birnbach, Germany) was utilized by an accredited skin cancer doctor for whole body skin checks for all participants. The system utilized a Medicam 1000 attachment which includes an integrated LED floodlight illumination and allowed up to 40 times magnification. The software version utilized with the mole mapping system with inbuilt AI was 3.4.1.0—(×64).

Following each participant's completion of the survey, the clinician completed the second half of the questionnaire, which included determination of Fitzpatrick skin type (*Gupta & Sharma, 2019*). The clinician then completed a whole-body skin examination where the number, type and location of suspected lesions were noted within the participant file using medical software (Xestro). Queried melanoma lesions included an automated predicted AI score which ranged from 0.00–0.20 "unsuspicious"; 0.21–0.49 "requiring clarification", 0.50–1.00 "suspicious, observe with high attention".

## Examination

All participants in this study completed the activity-specific survey and consented to a full-body skin check using the digital dermatoscope, where each participant was systematically examined on all skin surfaces, including the scalp, head, neck, torso, arms, palms, thighs, legs, soles, nails, and mucosae. Prior to the screening, each participant was asked if they had lesions of concern in concealed areas, and if so, an examination of those areas was also conducted. The clinician utilized the chaos and clues algorithm (*Rosendahl et al., 2012*) and prediction without pigment, a decision algorithm for non-pigmented skin malignancies (*Rosendahl et al., 2014*), as these screening methods have been shown to significantly increase the diagnostic accuracy of detecting AK, KC and MM when completed by an experienced clinician (*Dinnes et al., 2018*). Queried melanomas were captured by the Medicam, and an AI score was obtained and recorded in the participant's file. A predictive AI score was obtained on all pigmented lesions suspected of being a MM. Histopathology from a commercial laboratory was used as the ground truth (gold standard) and confirmed any suspected lesions following shave or punch biopsy.

## Artificial intelligence

Performance of AI was assessed by measuring predictive accuracy, sensitivity and specificity. A receiver operating characteristic (ROC) curve was created to graphically represent the performance of the binary classifier at different classification thresholds and quantify this performance *via* the area under the curve (AUC). True positive (TP) was classified as an AI predictive score of greater than 0.50 and histopathology confirmation of

a MM. True negative (TN) was classified as an AI value 0.49 or below and histopathology highlighting a benign lesion. A false positive (FP) was indicative as an AI value greater than 0.50 and negative histopathology, while a false negative (FN) was an AI value 0.49 or below and histopathology confirming a MM.

Accuracy, sensitivity and specificity were calculated from the following equations (*Baratloo et al., 2015*):

$$Accuracy = \frac{TP + TN}{TP + FP + FN + TN}$$

$$Sensitivity = \frac{TP}{TP + FN}$$

$$Specificity = \frac{TN}{TN + FP}.$$

Sensitivity in this study was recognized as the ability of the Medicam to correctly identify participants with a MM, which, based upon the manufacturer's recommendations is an AI score between 0.50 to 1.00 (inclusive) and is defined as suspicious-observe with high attention. Alternatively, specificity pertains to the ability of the Medicam to correctly identify participants without a MM, which, based upon the manufacturers recommendations, is an AI score between 0.00 to 0.20 (inclusive) and is defined as unsuspicious. Those AI scores between 0.21 to 0.49 (inclusive) are defined as unsuspicious-requiring clarification.

Test accuracy is defined as the ability to differentiate true positive and true negative cases from the population investigated. For this study, accuracy pertained to correctly identifying participants with or without the MM. *Hajian-Tilaki (2013)* reported that receiver operating characteristic (ROC) analysis is an analysis used in clinical epidemiology to quantify how accurately a medical device or system can discriminate between a diseased and non-diseased state (*i.e.*, diagnostic accuracy) (*Baratloo et al., 2015*). In addition, the ROC curve, a plot of sensitivity *vs.* specificity, was incorporated to remove diagnostic criteria as a confounding variable, and the area under the curve (AUC) a representation of test accuracy (*Hajian-Tilaki, 2013*).

## Statistical analysis

Kurtosis, skewness, Q-Q plots, and the Kolmogorov–Smirnov tests (with Lilliefors significance correction) were used to determine the normality of the data. Levene's inferential tests was used to evaluate heteroscedasticity. Excel (Microsoft Office 365; Microsoft Corporation, Redmond, WA, USA) and IBM's Statistical Package for Social Sciences (SPSS, Ver. 28.0) were used to conduct statistical analyses, which included demographics, independent sample t-tests, Chi-square tests and ANOVA (Bonferroni *post-hoc* test) to assess the significance of the relationship between the number of skin cancers and decade of age groups. The association between the frequency of skin malignancies, age, and body mass index was examined using a bivariate (two-tailed)

Pearson correlation coefficient. Alpha was predetermined to be significant between groups at $p < 0.05$.

The percentage of surfers, swimmers, or walkers/runners with an AK, BCC, SCC, IEC or MM was used to calculate point prevalence (*National Institute of Mental Health, 2022*). For example, the number of surfers identified with a specific skin cancer, multiplied by 100,000 and then divided by the total number of surfers, was used to obtain the standardized rate. A ratio of two sets of odds, such as the likelihood that an event would occure in one group (such as surfers) compared to another group (such as swimmers), is known as an odds ratio (OR). The odds in surfers were then divided by the odds in swimmers to arrive at the OR of surfers to swimmers (*Tenny & Hoffman, 2022*). This same process was completed to compare surfers with walkers/runners, and swimmers to walkers/runners respectively. All odds ratios were computed using the MedCalc statistical programme (https://www.medcalc.org/calc/odds_ratio.php) In the general Australian population, comparable standardised rates for AK, BCC, SCC, and MM were found in the literature (*Lai, Cranwell & Sinclair, 2018*; *Smith, 2019*; *Australian Institute of Health and Welfare, 2020*).

## RESULTS

### Participant's characteristics/demographics

A total of 423 participants (males $n = 208$, females $n = 215$; surfers $n = 108$, swimmers $n = 60$, walkers/runners $n = 255$) completed the survey and underwent a total body skin check for PSC, KC or MMs. Surfers were younger than both swimmers (46.16 *vs.* 53.38 yrs, $p = 0.010$) and walkers/runners (46.16 *vs.* 52.75 yrs, $p < 0.001$). Regarding body mass, surfers compared closely to swimmers (80.16 *vs.* 76.45 kg), however, surfers were heavier than walkers/runners (80.16 *vs.* 73.92 kg, $p = 0.001$). Similarly, surfers had a larger body surface area than walkers/runners (1.98 *vs.* 1.86 m$^2$, $p = 0.001$). All three groups reported similar body mass index (BMI) values (surfers 25.65 kg/m$^2$, swimmers 25.50 kg/m$^2$, walkers/runners 25.32 kg/m$^2$). The Fitzpatrick skin type between the groups did not reflect great variation, with the population consisting of mostly Fitzpatrick type 2 (fair, surfers 80.6%, swimmers 86.7%, walkers/runners 93.7%) and type 3 (light olive, surfers 19.4%, swimmers 13.3%, walkers/runners 6.3%) skin types.

In regard to outdoor sun exposure, swimmers and walkers/runners had greater number of year's exposure in their chosen recreational activities than surfers (swimmers 36.92 years, walkers/runners 34.75 years, surfers 25.98 years, $p < 0.001$). Walkers/runners had a greater number of weeks per year, hours per year and lifetime hours of sun exposure than the surfing and swimming groups (Table 1). More surfers spent significantly more time in the peak UVR period than both swimmers (91% *vs.* 75%, $p = 0.049$) and walkers/runners (91% *vs.* 53%, $p < 0.001$); surfers also experienced a significantly greater percentage of time in peak UVR than walkers/runners (32.59% *vs.* 21.77%, $p = 0.005$). More swimmers spent time during peak UVR than walkers/runners (75% *vs.* 53%, $p = 0.006$) and a greater percentage of time than walkers/runners (36.58% *vs.* 21.77%, $p = 0.002$).
**Table 1 Participant's demographics, values are mean/median (±SD), number or percent, [95% CI].** Specific *P* value included in parentheses where significant differences existed between groups.

| Parameter | Group (*n* = 423) | Surfers (*n* = 108) | Swimmers (*n* = 60) | Walkers/runners (*n* = 255) |
|---|---|---|---|---|
| Age (years) | 51.16/52.00 (15.5) | 46.16/44.50 (13.3) [43.6–48.7] | 53.38/54.50 (16.4) [49.2–57.6] [I]($p = 0.010$) | 52.75/55.00 (15.7) [50.8–54.8] [II]($p < 0.000541$) |
| Mass (kg) | 75.87/75.00 (15.3) | 80.16/80.25 (13.7) [77.6–82.8] | 76.45/78.00 (14.2) [72.8–80.1] | 73.92/72.00 (15.8) [72.0–75.9] [II]($p = 0.001$) |
| BMI (kg/m$^2$) | 25.50/25.00 (4.1) | 25.65/25.35 (3.5) [25.0–26.3] | 25.50/24.95 (3.6) [24.6–26.4] | 25.43/24.80 (4.5) [24.9–26.0] |
| Underweight (*n*) | 26 | 2 | 2 | 22 |
| Normal (*n*) | 188 | 49 | 29 | 110 |
| Overweight (*n*) | 158 | 46 | 24 | 88 |
| Obese (*n*) | 51 | 11 | 5 | 35 |
| Body surface area (m$^2$) | 1.90/1.91 (0.2) | 1.98/1.99 (0.2) [1.9–2.0] | 1.91/1.95 (0.2) [1.9–2.0] | 1.86/1.83 (0.2) [1.8–1.9] [II]($p < 0.000030$) |
| Experience (years) | 32.79/33.00 (16.6) | 25.98/25.00 (15.2) [23.1–28.9] | 36.92/36.50 (19.6) [32.0–42.3] [I]($p = 0.000079$) | 34.75/37.00 (15.7) [32.8–36.7] [II]($p = 0.000007$) |
| Hours/week | 4.96/4.00 (4.4) | 4.78/4.00 (4.6) [3.9–5.7] | 3.85/3.00 (3.4) [3.0–4.7] | 5.30/4.00 (4.6) [4.7–5.9] |
| Weeks/year | 42.39/52 (14.0) | 37.36/40.00 (14.7) [34.5–40.2] | 35.17/33.50 (14.9) [31.3–39.0] | 46.20/52.00 (12.0) [44.7–47.7] [II]($p < 0.001$ or $3.4515 \times 10^{-8}$) [III]($p < 0.001$ or $3.3107 \times 10^{-8}$) |
| Total hours/year | 226.16/156.00 (222.0) | 192.03/120.00 (208.45) [152.1–232.0] | 147.83/80.00 (167.0) [104.7–191.0] [II]($p = 0.001$) | 258.91/208.00 (232.5) [230.2–287.6] [III]($p = 0.018$) |
| Lifetime hours primary activity | 8,000/4,180 (9,926) | 5,594/2,475 (7,434) [4,169–7,019] | 5,686/2,600 (7,918) [3,641–7,732] | 9,554/6,160 (10,934) [8,206–10,902] [II]($p = 0.024$) [III]($p = 0.001$) |
| Activity during peak UV (%) | 66 | 91 [85–96] | 73 [62–85] [I]($p = 0.049$) | 53 [47–59] [II]($p < 0.001$ or $5.177 \times 10^{-12}$) [III]($p = 0.006$) |
| Activity percentage during peak UV (%) | 26.62/20.00 (30.4) | 32.59/30.00 (25.5) [27.7–37.5] | 36.58/30.00 (30.9) [27.8–45.4] | 21.77/5.00 (21.8) [18.0–25.5] [II]($p = 0.005$) [III]($p = 0.002$) |

**Notes:**
[I] Difference between surfers and swimmers.
[II] Difference between surfers and walkers/runners.
[III] Difference between swimmers and walkers/runners.
BMI, body mass index; BSA, body surface areas; UVR, Ultraviolet radiation.

**Table 2 Participant's prevention and screening demographics, values are percent.** Specific *P* value included in parentheses where significant differences existed between groups.

| Parameter | Group (*n* = 423) | Surfers (*n* = 108) | Swimmers (*n* = 60) | Walkers/runners (*n* = 255) |
|---|---|---|---|---|
| Uses any clothing prevention strategy (yes, %) | 86.5 | 80.6 | 65.0 | 94.1 [II]($p < 0.001$) [III]($p < 0.001$) |
| Surf hat, swim cap or hat (yes, %) | 61.0 | 32.4 | 23.3 | 82.0 [II]($p < 0.001$) [III]($p < 0.001$) |
| Uses rashie, wetsuit or t-shirt (%) | 75.7 | 76.9 | 50.0 [I]($p < 0.001$) | 81.2 [III]($p < 0.001$) |
| Use sunscreen (%) | 87.2 | 92.6 | 88.3 | 84.7 |
| Do you reapply sunscreen (%) | 46.3 | 50 | 61.7 | 41.2 [III]($p = 0.012$) |
| Use Zinc (%) | 35.2 | 78.7 | 20 [I]($p < 0.001$) | 20.4 [II]($p < 0.001$) |
| Sunscreen and Zinc (%) | 32.9 | 75 | 20 [I]($p < 0.001$) | 18 [II]($p < 0.001$) |
| Sunscreen or Zinc (%) | 89.8 | 96.3 | 90.0 | 87.1 [II]($p = 0.023$) |
| Previously underwent skin check (%) | 5 | 6.5 | 3.3 | 4.7 |
| Never | 5 | 6.5 | 3.3 | 4.7 |
| <6 months | 39.2 | 41.7 | 41.7 | 37.7 |
| 1 year ago | 29.8 | 27.8 | 31.7 | 30.2 |
| 2 years ago | 14.2 | 12.0 | 11.7 | 15.7 |
| 3 years ago | 4.5 | 7.4 | 5 | 3.1 |
| 4 years ago | <1 | 1.9 | 0 | <1 |
| 5 years ago | 1.2 | 1.9 | 1.7 | <1 |
| >5 years ago | 5.2 | <1 | 5 | 7.1 |
| Who performed last skin check (%) | 18.7 | 14.7 | 26.7 | 18.4 |
| GP | 18.7 | 14.7 | 26.7 | 18.4 |
| Skin cancer doctor | 73.7 | 79.4 | 60 | 74.7 |
| Dermatologist | 6.6 | 4.9 | 8.3 | 6.9 |
| Plastic surgeon | <1 | <1 | 5 | 0 |

**Notes:**
[I] Difference between surfers and swimmers.
[II] Difference between surfers and walkers/runners.
[III] Difference between swimmers and walkers/runners.

## Prevention and screening practices

Clothing protective strategies varied between the three groups, with walkers/runners significantly more likely to use a hat compared to surfers and swimmers (surfers 32.4%, swimmers 23.3%, walkers/runners 82.0%, *p* < 0.001). Swimmers were less likely to utilize upper body clothing compared to surfers (76.9% *vs.* 50.0%, *p* < 0.001) or walkers/runners (50.0% *vs.* 81.2%, *p* < 0.001). Sunscreen use did not vary greatly between the groups of participants, with most participants reporting applying sunscreen before sun exposure (surfers 92.6%, swimmers 88%, walkers/runners 84%). However, swimmers were significantly more likely to reapply sunscreen than walkers (61.7% *vs.* 41.2%, *p* = 0.012). Surfers were significantly more likely to apply zinc compared to swimmers and walkers/runners (78.7% *vs.* 20% and 20.4%, *p* < 0.001).

There was no significant difference identified between the groups regarding who conducted their last full skin examination, with most participants favouring a specialist

**Table 3 Participants' history related to skin cancer, values are number (*n*) or percent.** Specific *p* value included in parentheses where significant differences existed between groups.

| Parameter | Group (*n* = 423) | Surfers (*n* = 108) | Swimmers (*n* = 60) | Walkers/runners (*n* = 255) |
|---|---|---|---|---|
| Personal history of skin cancer (yes, %) | 46.6 | 44.4 | 50 | 46.6 |
| Family history of skin cancer (includes keratosis) (yes, %) | 50.1 | 57.4 | 51.7 | 46.6 |
| History of blistering sunburns as a child (yes, %) | 78.7 | 76.9 | 80 | 79.2 |
| Number of sunburns in previous 12 months (*n*) | 621 | 237 | 86 | 298 [II](*p* < 0.001) |
| Any lesions of concern (yes, %) | 46.6 | 51.9 | 38.3 | 46.3 |
| Personal history of skin cancer and skin cancer during screening (yes, %) | 40.2 | 40.7 | 45 | 38.8 |
| Skin cancer identified during screening (yes, %) | | | | |
| AK, KC, and MM | 61.9 | 65.7 | 55 | 62.0 |
| KC and MM only | 28.1 | 23.1 | 30 | 29.8 |

**Note:**
[II] Difference between surfers and walkers/runners.

general practitioner working exclusively in skin cancer medicine, followed by general practitioner, dermatologist, or plastic surgeon (Table 2). Most participants were self-referred (surfers 93.5%, swimmers 91.7%, walkers/runners 88.6%) and attended the skin clinic on either a 6-month or annual interval (69.5% surfers, 73.4% swimmers, 67.9% walkers/runners).

## History and risk related to skin cancer

Surfers, swimmers, and walkers/runners did not differ in their personal history of skin cancer, family history of skin cancer or experience of blistering sunburns before the age of 18 years old. When comparing the number of sunburns experienced over the past 12 months, surfers experienced more sunburns on average than swimmers (2.19 *vs.* 1.43) and significantly more sunburns than walkers/runners (2.19 *vs.* 1.17 *p* < 0.001). (Table 3).

## Skin cancer whole-body screening results

The skin specialist identified 167 participants (39.5%) with AK during the screening (surfers 47, swimmers 24, walkers/runners 96), 253 participants (59.8%) with KC (37 surfers, 45 swimmers, 171 walkers/runners) and 22 participants (5.2%) with MM (seven surfers, two swimmers, 13 walkers/runners). Surfers were more likely to have an AK identified during the skin examination than swimmers (43.5% *vs.* 40.0%) and with walkers/runners (43.5% *vs.* 37.6%). Surfers were twice as likely to have a MM than swimmers (6.48% *vs.* 3.33%, OR 2.01) and one and a half times more likely than walkers/runners (6.48% *vs.* 4.3%, OR 1.54). Conversely, surfers had fewer KC than both swimmers and surfers (Table 4).

Swimmers had a higher point prevalence than walkers/runners for BCCs (25% *vs.* 17.6%, OR 1.56), fewer SCCs (5% *vs.* 7.1%, OR 0.69) and marginally fewer IECs (8.3% *vs.* 8.6%, OR 0.96). When compared to the general Australian population, surfers, swimmers, and walkers/runners all had higher odds ratios which included BCCs (9.01, 16.22 11.45,
**Table 4 Skin cancer type by group.**

| Parameter | Point prevalence (no. persons, %) | | | Standardized rate per 100,000 | | | Odds ratios | Comparison of standardized rate to general population | | | |
|---|---|---|---|---|---|---|---|---|---|---|---|
| | Surfers | Swimmers | Walkers/runners | Surfers | Swimmers | Walkers/runners | | Australia general population | Surfers compared to Australia general population | Swimmers compared to Australia general population standardized rate | Walkers/runners compared to Australia general population standardized rate |
| Actinic keratosis | 43.5 | 40 | 37.6 | 43,519 [34,017–53,020] | 40,000 [27,238–52,762] | 37,647 [31,660–43,634] | [I]1.156 [0.609–2.195] [II]1.276 [0.808–2.015] [III]1.104 [0.621–1.963] | 40% (33) | 1.09 | 1.00 | – |
| BCC | 13.9 | 25 | 17.6 | 13,889 [8,007–21,623] | 25,000 [13,720–36,280] | 17,647 [12,936–22,358] | [I]0.484 [0.218–1.076] [II]0.753 [0.400–1.418] [III]1.556 [0.798–3.031] | 1,541 (33) | 9.01 | 16.22 | 11.45 |
| IEC | 3.7 | 8.3 | 8.6 | 3,704 [84–7,323] | 8,333 [1,133–15,533] | 8,627 [5,158–12,097] | [I]0.423 [0.109–1.640] [II]0.407 [0.137–1.212] [III]0.963 [0.349–2.655] | – | – | – | – |
| SCC | 2.8 | 5 | 7.1 | 2,778 [–372 to 5,927] | 8,333 [–678 to 10768] | 7,059 [3,894–10,224] | [I]0.543 [0.106–2.778] [II]0.376 [0.108–1.305] [III]0.693 [0.197–2.433] | 1,035 (33) | 2.68 | 8.05 | 6.82 |
| Melanoma | 6.48 | 3.33 | 4.3 | 6,481 [1,763–11,200] | 3,333 [–1,343 to 8,010] | 4,314 [1,803–6,824] | [I]2.010 [0.404–9.998] [II]1.537 [0.580–4.078] [III]0.765 [0.165–3.545] | 54.1 (11) | 119.80 | 61.61 | 79.74 |

**Notes:**

Values are percent (%), number (n) or [95% confidence interval]. BCC, basal cell carcinoma; IEC, intraepidermal carcinoma; SCC, squamous cell carcinoma; (1) *Lai, Cranwell & Sinclair (2018)*; (2) *Australian Institute of Health and Welfare (2022)*.

[I] Odds ratio between surfers and swimmers.

[II] Odds ratio between surfers and walkers/runners.

[III] Odds ratio between swimmers and walkers/runners.

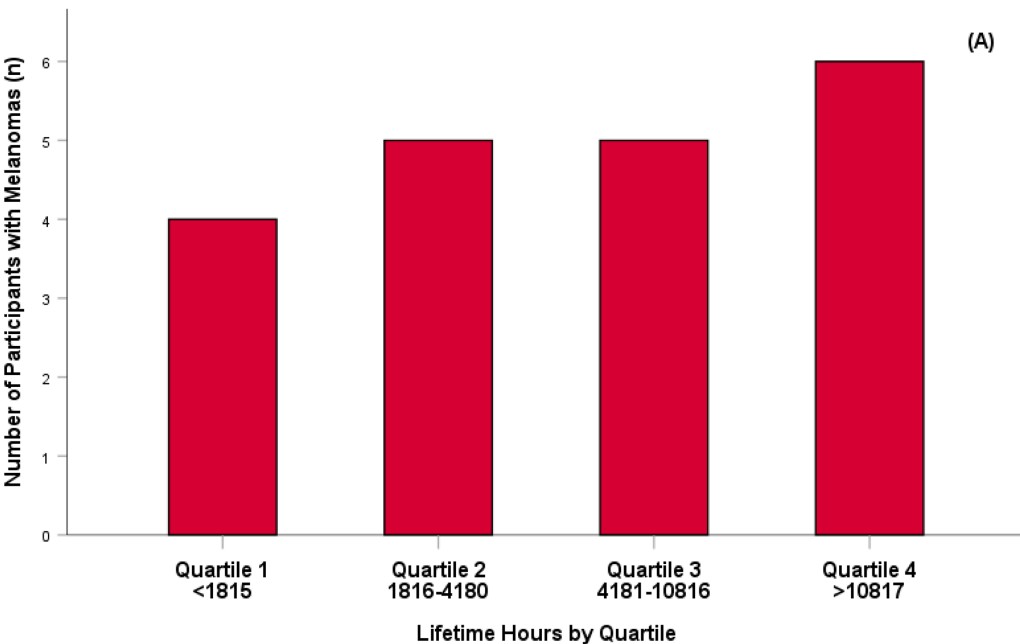

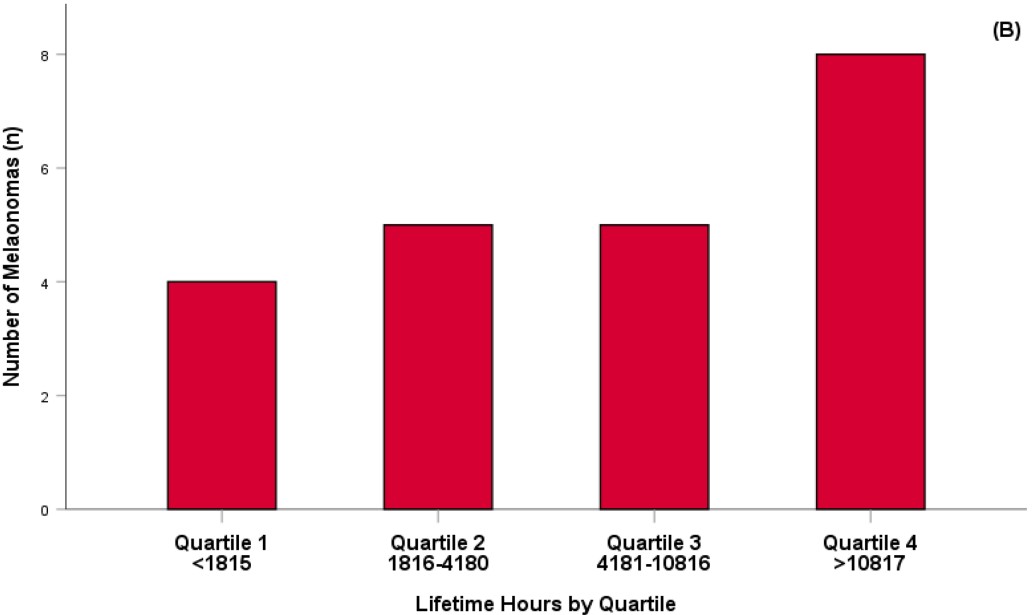

**Figure 1 Number of participants identified with a melanoma skin cancer per quartile (A) and number of melanoma skin cancer identified per quartile (B).**

respectively), SCCs (2.68, 8.05, 6.82, respectively) and MM (119.8, 61.61, 79.74, respectively).

When the population was combined and their estimated lifetime UVR exposure (accumulated hours per year multiplied by the number of years completing activity) was

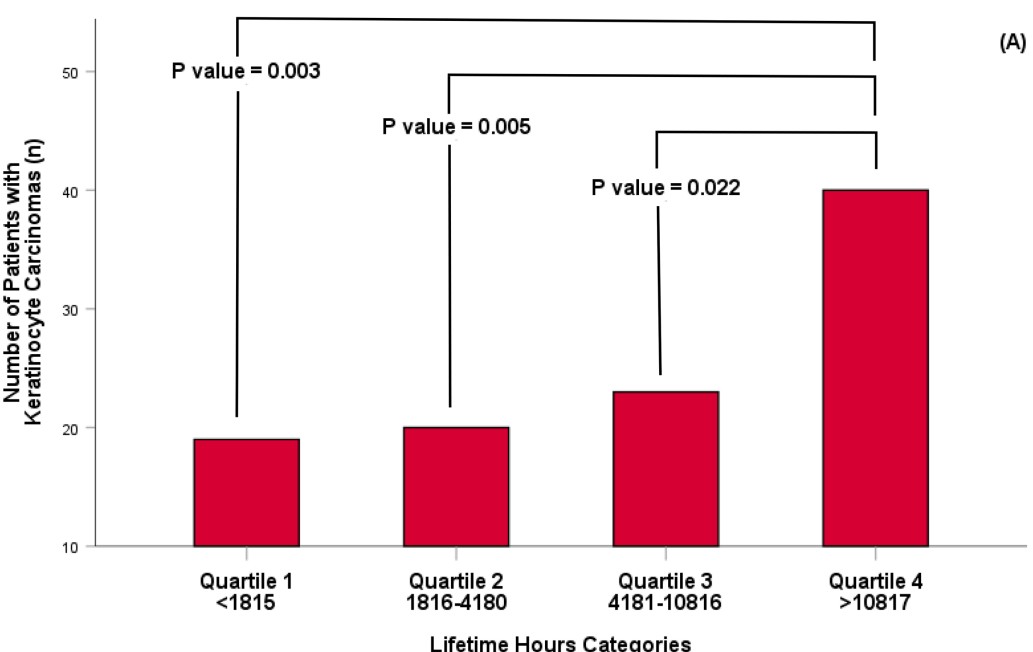

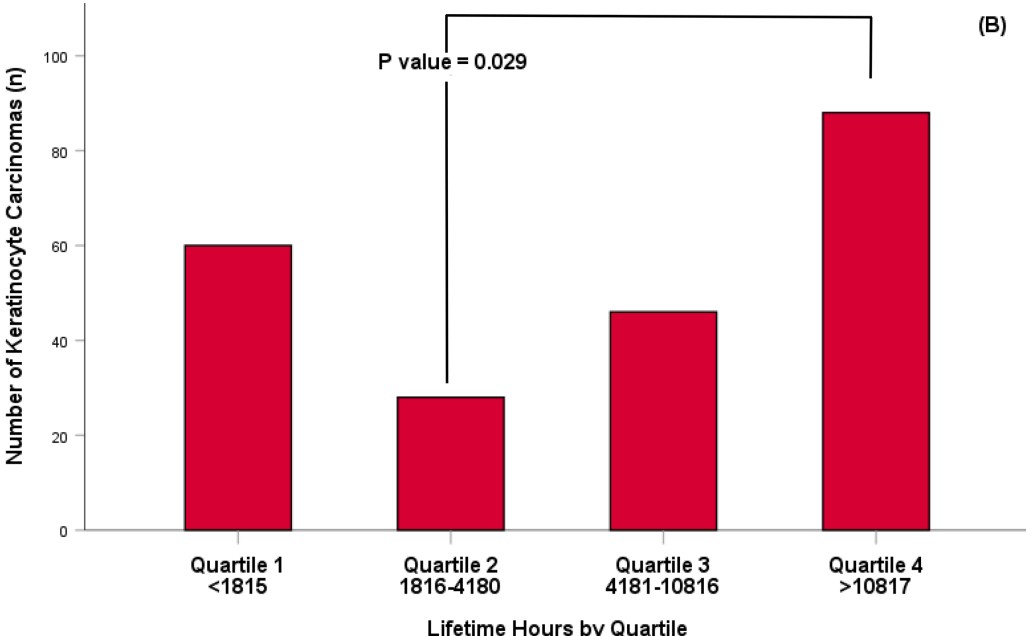

**Figure 2 Number of participants with a keratinocyte carcinoma per quartile (A) and number of keratinocyte carcinoma identified per quartile (B).**

expressed as quartiles, there was no significant difference for the number of participants diagnosed with a MM (Fig. 1A) or the number of MM diagnosed in the population (Fig. 1B). Conversely, there was a significant difference between the number of individuals who had a KC diagnosed and the lifetime hours of UVR exposure experienced between

participants in quartile four and participants in each preceding quartile (quartile 1; $p = 0.003$, quartile 2; $p = 0.005$, quartile 3; $p = 0.022$) respectively (Fig. 2A). Lifetime UVR exposure and the total number of KC diagnosed was significantly different for participants in Quartile 2 and 4 ($p = 0.029$) and no significant difference between the remaining quartiles (Fig. 2B).

## Fitzpatrick skin type and skin cancer

Participants were classified as either Fitzpatrick skin type 2 (88.2%) or skin type 3 (11.8%). There were no participants identified as skin type 1, 4, 5 or type 6. Regarding primary activity, surfers (80.6% *vs.* 19.4%), swimmers (86.7% *vs.* 13.3%) and walkers/runners (93.7% *vs.* 6.3%) were identified as type 2 or type 3 respectively. Coincidently, a greater amount of KC and MM were detected in type 2 skin types (89.4%) compared to type 3 skin types (10.6%).

## Gender differences

Primary activities had clear gender tendencies, with males more likely to identify as surfers (males 77.8% *vs.* females 22.2%), swimmers were split evenly across the genders (males 45.0% *vs.* females 55.0%) and more walking/running participants were female (males 38.0% *vs.* females 62.0%). Males spent significantly longer activity time exposed to the sun than females during their average activity session (males 73.5 min *vs.* females 61.7 min, $p = 0.011$). Conversely, females spent significantly more weeks completing their chosen activity, on average, than males (males 40.5 weeks *vs.* females 44.2 weeks, $p = 0.006$). There was no significant difference between primary activity lifetime hours between the genders, though males reported a greater amount (males 8,867.7 h *vs.* females 7,177.2 h, $p = 0.086$).

Significantly more males were diagnosed with a skin cancer than females (males 28.4% *vs.* females 20.0%, $p = 0.045$) which was reflected in a greater number of overall KC identified (males 149 *vs.* females 73, $p = 0.010$). No significant finding between the genders was identified for MM though males had a greater number than females (males, $n = 13$ *vs.* females, $n = 7$). Significantly more male participants were registered at least one AK than female participants (males 47.1% *vs.* females 32.1%, $p = 0.002$) during their skin examination.

## Artificial intelligence: characteristics of queried lesions

Of the 423 participants, 48 lesions were identified clinically as suspicious of MM during skin examination and a predictive AI score was subsequently obtained. Histopathology confirmed 15 MM and 33 benign pigmented lesions. The subtypes of MM were classified as either superficial spreading ($n = 1$), lentigo maligna ($n = 5$), lentiginous ($n = 3$), not otherwise specified ($n = 6$) and were classified as *in-situ* ($n = 4$), invasive ($n = 1$) or both ($n = 10$) (Table 5). In regard to the benign lesions, 25 out of 33 lesions (75.8%) were confirmed *via* histopathology to be a naevus derivative, three were classified as pigmented seborrheic keratosis, two dermatofibroma and a single lesion classified as an

**Table 5 Confirmed AI positive melanoma characteristics.**

| Characteristics | SSM | LMM | Lentiginous | NOS | Total |
|---|---|---|---|---|---|
| Age (mean (±SD)) | 38 | 65.4 (16.3) | 66.00 (3.6) | 45.7 (10.4) | 55.8 (15.4) |
| Female (*n*, %) | 1, 100 | 1, 20 | 1, 33.3 | 3, 50 | 6, 40 |
| Male (*n*, %) | 0, 0 | 4, 80 | 2, 66.7 | 3, 50 | 9, 60 |
| Localisation (*n*, %) | | | | | |
| Scalp/face | – | – | – | – | – |
| Trunk/extremities | 1, 100 | 5, 100 | 3, 100 | 6, 100 | 15, 100 |
| Invasiveness | | | | | |
| *In situ* (*n*, %) | 1, 100 | 5, 100 | 3, 100 | 5, 83.3 | 14, 93.3 |
| Invasive (*n*, %) | 1, 100 | 3, 100 | 3, 100 | 4, 66.7 | 11, 73.3 |
| Breslow thickness (mean (±SD)) | 0.40 | – | 0.40 (0.28) | 0.87 (0.98) | 0.63 (0.69) |
| Clark level (*n*, %) | | | | | |
| Level1 | 0, 0 | 5, 100 | 1, 33.3 | 3, 50 | 9, 60 |
| Level2 | 0, 0 | 0, 0 | 2, 66.7 | 3, 50 | 5, 33.3 |
| Level3 | 1, 100 | 0, 0 | 0, 0 | 0, 0 | 1, 6.7 |
| AI characteristics | | | | | |
| AI score (mean (±SD)) | 0.12 | 0.72 (0.33) | 0.68 (0.31) | 0.40 (0.33) | 0.54 (0.34) |
| AI size mm$^2$ (mean (±SD)) | 30 | 92.8 (16.1) | 113.3 (61.1) | 28.7 (34.5) | 67.1 (49.6) |

Note:
SSM, superficial spreading melanoma; LMM, lentigo maligna melanoma; NOS, not otherwise specified.

**Table 6 Benign lesions queried for melanoma.**

| Characteristics | Naevus | Pigmented seborrheic keratosis | Solar keratosis/solar lentigo | IEC | Dermatofibroma | Foreign body | Total |
|---|---|---|---|---|---|---|---|
| Age (mean (±SD)) | 45.1 (14.9) | 49.3 (11.9) | 68.0 | 78.0 | 55.0 (2.8) | 52.0 | 48.0 (15.0) |
| Female (*n*, %) | 8, 32.0 | 1, 33.3 | 1, 100 | 1, 100 | 1, 50 | 0, 0 | 12, 36.36 |
| Male (*n*, %) | 17, 68.0 | 2, 66.7 | 0, 0 | 0, 0 | 1, 50 | 1, 100 | 21, 63.64 |
| Localisation (*n*, %) | | | | | | | |
| Scalp/face | 1, 4 | 0, 0 | 0, 0 | 1, 100 | 1, 50 | 0, 0 | 3, 9.1 |
| Trunk/extremities | 24, 96 | 3, 100 | 1, 100 | 0, 0 | 1, 50 | 1, 100 | 30, 90.9 |
| AI characteristics | | | | | | | |
| AI score (mean (±SD)) | 0.48 (0.33) | 0.17 (0.04) | 0.85 | 0.44 | 0.5 (0.54) | 0.93 | 0.47 (0.33) |
| AI size mm$^2$ (mean (±SD)) | 24.2 (24.3) | 35.7 (17.4) | 16.0 | 56.0 | 92.0 (39.6) | 18.0 | 29.9 (28.5) |

Note:
AI, artificial intelligence; IEC, intraepidermal carcinoma.

intra-epidermal carcinoma (IEC), actinic keratosis arising with solar lentigo and foreign body respectively (Table 6).

The mean AI score of the 48 lesions was 0.50 (±0.34) with a range of 0.10 through to 0.93. The mean area of the lesions was 41.52 mm$^2$ (±39.84) with a range of 4.0 to 180 mm$^2$.

**Table 7 Artificial intelligence score classification *vs*. histopathology.**

| | | Histopathology | | |
| | | Positive | Negative | Total |
|---|---|---|---|---|
| Score classification | Unsuspicious 0–0.20 (*n*, %) | 5 (35.3%) | 13 (38.7%) | 18 (37.5%) |
| | Requires classification 0.21–0.49 (*n*, %) | 2 (11.8%) | 5 (16.7%) | 7 (14.6%) |
| | Suspicious 0.50–1.00 (*n*, %) | 8 (52.9%) | 15 (45.2%) | 23 (47.9%) |
| Total | | 15 | 33 | 48 |

**Table 8 Artificial intelligence score *vs*. histopathology.**

| | | Histopathology | | |
| | | Positive | Negative | Total |
|---|---|---|---|---|
| AI score diagnostic | Positive (>0.50) | 8 (53.3%) | 15 (45.5%) | 23 (47.9%) |
| | Negative (<0.49) | 7 (46.7%) | 18 (54.5%) | 25 (52.1%) |
| Total | | 15 (100.0%) | 33 (100%) | 48 (100%) |
| Sensitivity (%) | 53.33 | | | |
| Specificity (%) | 54.44 | | | |
| Accuracy (%) | 54.17 | | | |

Queried lesions were more likely to be found on the back followed by legs, chest, abdomen and arms (*n* = 30) than on the face, neck or scalp (*n* = 3, *p* = 0.083).

### Artificial intelligence: diagnostic accuracy of queried lesions

Of the 48 queried lesions, the AI score classification, based upon the technical manual, was classified as 23 suspicious lesions (AI score between 0.50 and 1.0 out of 1.00), seven lesions that warrant further clarification (AI score between 0.21 and 0.49 out of 1.00) and 18 unsuspicious lesions (AI score between 0.00 and 0.49 out of 1.00) (Table 7). When crosstabulation of histopathology and suspicious lesions, with a cut off of greater than 0.50, suspected lesions were classified as either true positive (*n* = 8), true negative (*n* = 18), false positive (*n* = 15) and false negative (*n* = 7) respectively (Table 8). Sensitivity, specificity, and accuracy were calculated from the queried lesions as 53.33%, 54.44% and 54.17% respectively. The AUC of ROC was calculated as 0.540 (Fig. 3). The area of histopathology confirmed MM reflected a positive correlation ($r^2$ = 0.301) and non-melanocytic lesions minimal correlation ($r^2$ = 0.007) (Fig. 4).

## DISCUSSION

In this study we aimed to identify the point prevalence of skin cancer in outdoor (walkers or runners) and aquatic (surfers or swimmers) enthusiasts. A total of 423 participants completed a survey and underwent a full body skin examination by a skin cancer doctor representing the largest skin cancer screening study to date. This study also contributes to the limited literature available on the prevalence of skin cancer in walkers, runners,

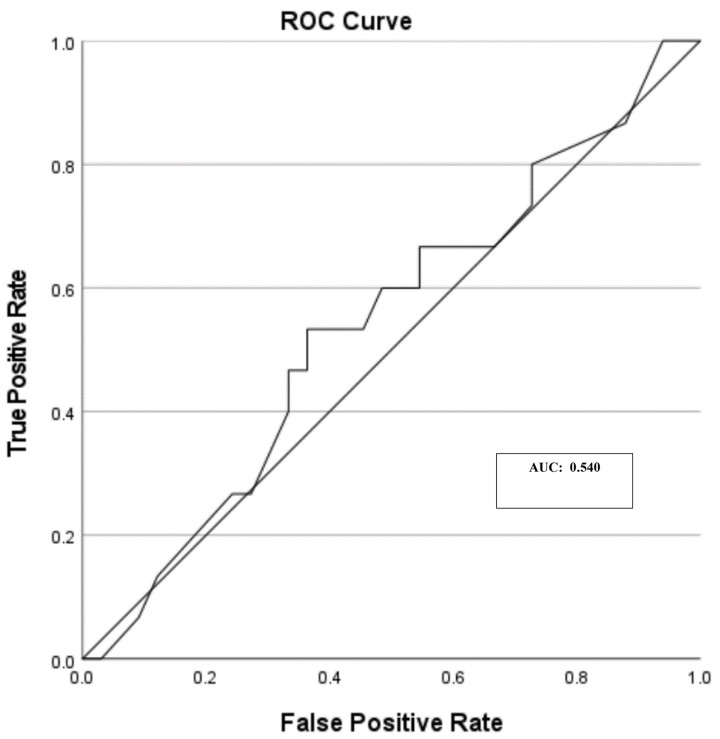

**Figure 3 Queried melanoma skin lesions (*n* = 48) and area under the curve (AUC).**

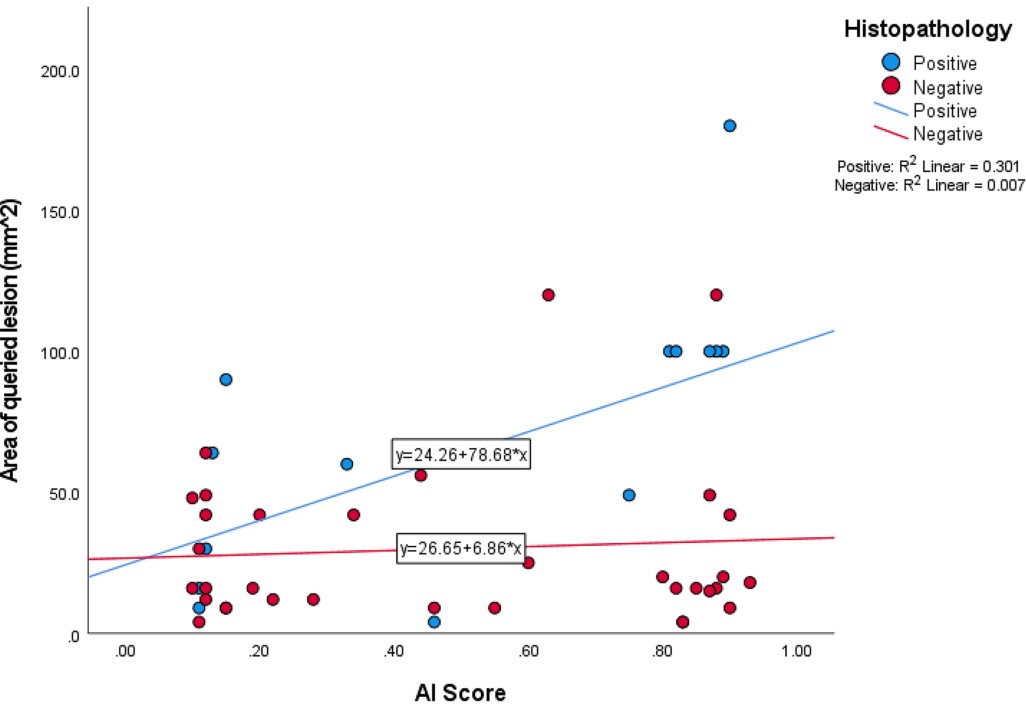

**Figure 4 Queried AI scores *vs.* size of queried lesion.**

swimmers and surfers in Australia. Our findings identified a high lifetime prevalence of skin cancer in the groups investigated compared to the Australian general population. Our findings for AK were similar to that previously reported in the literature. However, our findings for BCC, SCC and melanoma were much higher for all groups in comparison to the general Australian population (*Lai, Cranwell & Sinclair, 2018*; *Smith, 2019*; *Australian Institute of Health and Welfare, 2020*). With lifetime prevalence expressed as standardized rates (per 100,000), surfers had the highest rates for AK and melanoma. Swimmers were identified as having the highest age standardized rates for BCC and SCC while walkers had the highest age standardized rate for IEC. Given our participants had reasonably high UVR exposure, with high percentage (53% to 91%) during peak UVR periods, this is not a surprising finding and has previously been reported by our group (*Climstein et al., 2022*).

The surfing population from our study was twice as likely to have a MM than swimmers and 1.5 times as likely than walkers/runners. Although surfers spent fewer lifetime hours outside exposed to UVR as part of their recreation than the other populations investigated, surfers spent longer periods outdoors during peak UV exposure. To help understand why surfers might experience a higher prevalence of MM, we investigated if reflectance of UVR from water was a contributing factor. The *World Health Organisation (2016)* has reported ground reflection of UVR from grass/soil/water is 10%, dry sand 15% and sea foam is 25%, providing further rationale for the higher prevalence of MM in surfers.

There have been limited skin cancer studies, survey and screening, of surfers. *Climstein et al. (2016)* conducted a screening study of 1,348 recreational ($n = 767$) and competitive ($n = 581$) Australian surfers and reported their rates of skin cancer (as a group) to be much lower than we observed in our current study. Basal cell carcinoma was the most common KC reported (6.8%) followed by melanoma (1.4%) and SCC (0.6%). *Climstein et al. (2016)* did report UVR exposure and their weekly exposure for their group (6.7 hrs/wk ± 5.6) was approximately 35% higher than our three groups combined (4.96 hrs/wk ± 4.0).

With regard to skin cancer screening studies and surfers, *Dozier et al. (1997)* conducted the first screening study on surfers. They screened 110 surfers competing at a surfing contest in Texas (USA). Approximately one-third (32.7%) of the surfers were diagnosed with AKs and BCCs. The authors stated there were no lesions suggestive of SCC or MM. Their prevalence of AK was lower than we observed for our three groups (walkers/runners 37.6%; swimmers 40%; surfers 43.5%). However, their prevalence of BCC was lower than we observed (surfers 13.9%; walkers/runners 17.6%; swimmers 25%). *Dozier et al. (1997)* did not report exposure to UVR in their participants, however they did report that only 33% of their participants utilized sunscreen, compared to our participants, who reported a much higher rate (84.7% (walkers/runners) to 92.6% (surfers)).

*Climstein et al. (2022)* recently published an analysis of skin cancer rates in 171 surfers and 55 swimmers. The highest lifetime point prevalence for AK was demonstrated in surfers (37.1%) as opposed to swimmers (21.8%). These values are slightly lower than we observed in our present study. *Climstein et al. (2022)* also reported BCCs and found approximately 12.5% of their participants reported having a BCC, which is lower than we observed in our current study (swimmers 25%, walkers/runners 17.6%). The prevalence of

SCC in both studies was similar. In both studies the prevalence of MM was highest in surfers, 5.2% in surfers and 1.8% in swimmers whereas in our current study our prevalence of MM ranged from a low of 3.3% (swimmers) to a high of 6.48% (surfers). Although our current study had a greater number of participants compared with *Climstein et al. (2022)* (423 *vs.* 171), the lifetime aquatic or non-aquatic (walkers and runners) hours of UVR exposure was somewhat similar between studies in surfers (approx. 8,000 h) whereas swimmers in the current study had approximately one-half estimated lifetime UVR exposure (9,134 *vs.* 5,594 h). This is an interesting finding as the participants in our current study were, on average, older than the *Climstein et al. (2016)* survey participants. The activity completed during peak UVR was also somewhat lower in our current study.

*Ambros-Rudolph et al. (2006)* in Austria screening 210 marathon runners, found no MM in their participants, whereas 4.3% of our walkers/runners were diagnosed with MM. *Ambros-Rudolph et al. (2006)* also reported 1.9% of their participants had KC, however, they did not differentiate between BCC or SCC. In our current study, 17.6% of our walkers/ runners were identified with BCCs and 7.1% with SCCs. Additionally, *Ambros-Rudolph et al. (2006)* stated that the histopathology results were not available due to Austrian privacy laws.

*Climstein et al. (2022)* previously investigated the lifetime prevalence of skin cancer in Australian swimmers ($n$ = 55). However, at present, there were no other studies identified in the literature which have investigated this cohort profile worldwide. In that study from *Climstein et al. (2022)* the authors reported the point prevalence for swimmers as AK (21.8%), BCC, (14.5%), SCC (3.6%) and MM (1.8%). Our current study had a higher point prevalence in swimmers where AKs were approximately 2-fold higher (40%), BCCs slightly higher (+21.4%), SCCs 2-fold higher (7.1%) and MM 85% higher (3.3%). The mean age of the swimmers in both studies was similar (42.7 *vs.* 46.2 yrs) however, the estimated lifetime activity of swimming was approximately 60% higher in the earlier study swimming participants (*Climstein et al., 2016*). *Nelemans et al. (1994)* speculated that recreational exposure to UVR may not be the only mechanism for the development of MM. The authors hypothesized that carcinogens in the water may play a role in the development of MM. In their study they compared 128 MM patients to 168 patients who has never swam. Results revealed an odds ratio for MM of 2.2 in the patients who had swam prior to the age of 15 to those who had never swam. *Dennis et al. (2008)* conducted a systematic review and identified that sunburns were also a mechanism for MM development. In our present study, 4.0% of our participants (as a group) who reported blistering sunburns as a child were found to have a MM, of those, only two were swimmers. Whereas only 1.4% of our participants who had reported ever using a tanning bed were found to have a MM, of those one was a swimmer.

With regard to other outdoor activities, our prevalence of both AK and KC reported was much higher than that reported in skiers and ski workers. *Gilaberte et al. (2020)* surveyed 219 ski employees who only reported AKs in their participants and their prevalence was significantly lower (14.6%) that the rates of AKs in our walkers/runners (37.6%) up to 40% in swimmers. *Snyder et al. (2020)* screened ski workers and they reported a similar prevalence of AK (35.4%) as in our study. However, their rates of BCC (6.4%) and SCC

(1.6%) were well below our rates of 13.9% in surfers and a high of 17.6% in walkers/ runners. Neither study (*Gilaberte et al., 2020*; *Snyder et al., 2020*) identified any MM in their participants.

*del Boz et al. (2015)* screened 200 golfers and 151 golf course workers in Spain. Similar to *Gilaberte et al. (2020)* they only identified AKs in their participants with no other KC or MM skin cancers identified. Approximately 40% of the golfers were identified with AKs whereas only 10% of golf workers were identified with AK. *Noble-Jerks, Weatherby & Meir (2006)* conducted a survey on 164 retired Australian cricket players (mean age 45.2 yrs ± 12.1), they reported that 38% of the cricketers reported at least one skin cancer however, their study did not report the specific type of skin cancer as its focus was on skin cancer protection strategies.

Our findings for skin cancer examinations indicates the majority (68–74%) of our participants underwent regular skin screenings whereas *De Castro-Maqueda et al. (2020)* found in elite aquatic athletes (surfers, windsurfers, sailors) that the majority (84%) had not had a skin check-up and a similar percent (88%) did not conduct self-examinations for suspicious moles or spots. Previous studies by De Castro-Maqueda have brought attention to sun exposure in elite kite surfers (*De Castro-Maqueda et al., 2020*, *2021*). The geographic locations for these studies (Spain) and here in Australia may partly explain some disparity in findings. Australia has long been recognized to have the highest skin cancer incidence in the world. Alongside a media campaign aimed at reducing the incidence of skin cancer in Australia, we speculate the Australian public are well informed that abnormal lesion on their skin warrants a skin examination by a medical professional (general practitioner or specialist).

Our findings with regard to the sensitivity, specificity and accuracy of AI in MM risk were well below values that was recently reported in the literature. *Jones et al. (2022)* completed a systematic review of artificial intelligence and machine learning algorithms in the early detection of melanoma and other skin cancers in the community and primary care setting. Their review included 197 studies which reported the sensitivity, specificity and accuracy for melanoma detection. Overall, they reported a mean sensitivity of 84.2% (range 81.6–86.8%), mean specificity of 89.1% (range 87.1–91.0%) and an accuracy mean of 89.5% (range 88.2–90.8%).

Specific to the technology we utilized in this study (FotoFinder Vexia), *MacLellan et al. (2021)* investigated the sensitivity and specificity of three commercial noninvasive imaging devices, one of which was identical to that employed in our study. *MacLellan et al. (2021)* reported the FotoFinder to have superior sensitivity (88.1%) and specificity (78.8%) that the other commercial imaging devices from dermoscopic images. However, *MacLellan et al. (2021)* findings are much higher than we observed, with a sensitivity of 53.5% and specificity of 54.4%. The FotoFinder reports (*FotoFinder Systems GmbH, 2020*) using a deep-learning algorithm and complex machine learning to continually update and ultimately increase the sensitivity and specificity. *MacLellan et al. (2021)* does conclude that the FotoFinder had not been tested in a clinical setting, such as in our study. The FotoFinder manual (*FotoFinder Systems GmbH, 2020*) does state that it is one of the first systems to use a conventional neural network for skin cancer examinations. We had

expected to find a sensitivity and specificity equal to, or better than that previously reported by *MacLellan et al. (2021)*.

Given our findings with regard to sensitivity, specificity and accuracy, we sought consultation with highly experienced researchers who regularly use noninvasive imaging devices incorporating AI in skin cancer research. They speculated that as the FotoFinder technology was developed in Germany, that the company trained and tested their algorithm on non-sun damaged skin, and this may account for the lower-than-expected sensitivity, specificity and accuracy we identified in this study. At present, no studies have been completed which compared MM detection *via* AI in sun damaged *vs.* non-sun damaged skin. Since the time of writing, FotoFinder has been adjusting its CNN to address this issue, and additional images from participants with sun damaged skin are being provided to the company to improve the next iteration of its AI software.

### Study limitations

The authors note several limitations of this study. First, the present study had a relatively small sample size of AI queried lesions. Higher numbers zutilizing the FotoFinder technology have been reported by *Winkler et al. (2020)*, *Haenssle et al. (2020)* and *Sies et al. (2020)*. However, these studies focussed upon image datasets to compare the performance of the technology to specialists. To the authors knowledge, the present study is the first to test this technology within an Australian population within a clinical, primary care setting.

Secondly, one specialist identified the lesions of interest, which inevitably poses a selection bias for the current study. Thirdly, the FotoFinder could not capture all lesions of interest, leading to bias. Capture limitations included tattoo pigmentations, and large or irregularly shaped lesions outside the dermatoscope capture ability, such as on the helix and antihelix of the ear. Improved lenses and camera attachments for imaging hard to reach places have been developed and this can be addressed in future studies. These factors were anomalies and could be considered outliers however, should be highlighted for the sake of transparency.

### CONCLUSIONS

Exposure to UVR and skin cancer continue to be major health issues in Australia and the current study highlights the clinical importance of regular skin screening, especially in surfers and swimmers alongside individuals who spend significant time exposed to UVR while walking or running outdoors. Rates of KC and MM were notably higher in aquatic and land-based enthusiasts compared to the reported figures of the general Australian population. These findings further highlight the clinical importance of sun-safe protection measures and regular skin screening in aquatic and non-aquatic persons who spend significant hours outdoors. When compared to the general Australian population, surfers had the highest OR for MM (OR 119.8). Factors including physical and chemical sun protection, seeking shade where possible, and limiting activities during the middle of the day may be helpful in limiting an individual's likelihood of developing skin cancer.

The emerging use of AI technology in dermatology may provide clinicians with a useful tool for the early detection of MM. Importantly, the lower-than-expected sensitivity and

specificity of the AI we observed in this study may be attributed to participants having a much higher prevalence of severely sun-damaged skin than the European or American datasets used to train the AI.

The clinical role of AI as a diagnostic or adjunctive tool is still evolving and is promising. While efforts are currently underway to incorporate sun damaged skin and images of small melanomas into the CNNs. Until this is refined further, this study suggests that clinical caution needs to be taken when using AI for individuals with potentially sun damaged skin. AI is perhaps best used in high UV environments as an adjunctive rather than a diagnostic alternative to experienced clinical acumen.

## ACKNOWLEDGEMENTS

We would like to express our appreciation for all the participants who took part in our study and took the time to share their experiences. We would also like to acknowledge the input from Professor Peter Soyer and Professor Monika Janda for their invaluable input into our AI findings. We would also like to extend our sincere thanks to Professor Pat O'Shea, friend and mentor, for instilling a passion for research; you are sincerely missed but not forgotten.

### Funding
The authors received funding from Johnson and Johnson for the purchase of the high-resolution digital dermatoscope with artificial intelligence which enabled the study to be conducted. The funders had no role in study design, data collection and analysis, decision to publish, or preparation of the manuscript.

### Grant Disclosures
The following grant information was disclosed by the authors:
Johnson and Johnson.

### Competing Interests
Mike Climstein is a Section Editor for PeerJ (Sports Medicine and Rehabilitation). Michael Stapelberg is self-employed operating at John Flynn Hospital Specialist Centre. Jeremy Hudson and Paul Coxon are employed at North Queensland Skin Centre.

### Author Contributions
- Ian J. Miller conceived and designed the experiments, performed the experiments, analyzed the data, prepared figures and/or tables, authored or reviewed drafts of the article, and approved the final draft.
- Michael Stapelberg conceived and designed the experiments, performed the experiments, authored or reviewed drafts of the article, and approved the final draft.
- Nedeljka Rosic conceived and designed the experiments, analyzed the data, authored or reviewed drafts of the article, and approved the final draft.

- Jeremy Hudson analyzed the data, authored or reviewed drafts of the article, and approved the final draft.
- Paul Coxon analyzed the data, authored or reviewed drafts of the article, and approved the final draft.
- James Furness analyzed the data, authored or reviewed drafts of the article, and approved the final draft.
- Joe Walsh analyzed the data, prepared figures and/or tables, authored or reviewed drafts of the article, and approved the final draft.
- Mike Climstein conceived and designed the experiments, performed the experiments, analyzed the data, prepared figures and/or tables, authored or reviewed drafts of the article, and approved the final draft.

### Data Availability

The raw measurements are available in the Supplemental Files.

### Supplemental Information

Supplemental information for this article can be found online at http://dx.doi.org/10.7717/peerj.15737#supplemental-information.

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
