# Peer review of "Implementation of artificial intelligence for the detection of cutaneous melanoma within a primary care setting: prevalence and types of skin cancer in outdoor enthusiasts"

_PeerJ, doi:10.7717/peerj.15737_

## Round 0.1 · original submission · Minor Revisions

The research presented aimed at assessing skin cancer incidence in Australian surfers, swimmers, walkers/runners. A commercial mole mapping system with in-built artificial intelligence was used to screen the skin of participants. Results show that keratinocyte and melanoma carcinomas are higher in aquatic and land-based sports people compared to reported data of the general Australian population.

Two reviewers find the paper acceptable with minor revision. I agree with this assessment and ask the authors to consider the comments of the reviewers in their revision.

Reviewer 1 ·

Basic reporting

The paper discusses the use of artificial intelligence (AI) in detecting skin cancer in its early stages. The authors report that AI has shown promising results in detecting melanomas from dermoscopic image sets, with sensitivity and specificity values equivalent to specialists. The study aims to assess the incidence of skin cancer in individuals who regularly participate in outdoor activities in Australia, where melanoma rates are high. The authors conducted a cross-sectional study that included a survey, total body skin cancer screening, and the use of commercially available AI software (FotoFinder) to assess the predictive risk of melanoma development. The study found that surfers had the highest prevalence and odds ratio for melanoma, followed by walkers/runners and swimmers. The AI software had a sensitivity of 53.33%, specificity of 54.44%, and accuracy of 54.17%. The authors highlight the importance of sun-safe protection measures and regular skin screening for individuals who spend significant hours outdoors. They caution against expecting AI to substitute examination by a clinician, especially with patients with sun-damaged skin.

Overall, the manuscript is fascinating because it draws some attention to outdoor activities for those who experience high levels of intermittent ultraviolet radiation (UVR), which is strongly associated with developing different types of skin cancer, including melanoma.

The quality of the figures needs to be edited and improved.

Experimental design

The study falls within the Scope of the journal. The research question is well-defined, relevant and meaningful. The method is described with sufficient detail & information to replicate.

Validity of the findings

The study reports results that can be considered a meaningful replication, as the rationale and benefits to the literature are clearly stated. All underlying data have been provided, and they are robust and statistically sound. The conclusions are well-stated and linked to the original research question.

Additional comments

Here are some comments that can potentially improve this study:
1. The authors do not provide details on the training and validation of the AI software used in the study. Authors should include a more detailed description of the AI software, including the data used for training and validation, to evaluate its performance better.
2. The sample size for some of the subgroups (such as swimmers) was small, which limits the ability to draw meaningful conclusions about the incidence of skin cancer in these groups. Future studies should consider recruiting larger sample sizes for each subgroup to ensure more accurate estimates of skin cancer incidence.
3. The authors stated that "the findings with regard to sensitivity, specificity and accuracy, we sought consultation with highly experienced researchers who regularly use non-invasive imaging devices incorporating AI in skin cancer research. They speculated that as the FotoFinder technology was developed in Germany, the company trained and tested their algorithm on non-sun-damaged skin, and this would account for the lower-than-expected sensitivity, specificity and accuracy". Could the authors consider using their own data to re-train the FotoFinder technology if they have proper authorization for re-training this technology? This will change these results much better.
4. The resolution of the figures is insufficient and requires editing.

Reviewer 2 ·

Basic reporting

the authors use clear and scientific English throughout the writing of the article.
Some interesting reference has been attached to this study but in general it is sufficient and valid
The structure, figures and tables are correct
contains relevant and interesting results for publication.

Experimental design

The research is original and valid for publication in this journal
Research question well defined, relevant & meaningful.
The research was conducted in conformity with the prevaling ethical standars in this field
Methods described with sufficient detail. however, you are suggested to specify more of the questionnaire used.

Validity of the findings

A clear justification is established that adds novelty to the existing literature
The data es corrected the statistics are robust and correct
The conclusiion is connected to othe original question investigated and supported by results

Additional comments

The article is interesting and represents an advance for the study of skin cancer and the aquatic sports. H the statistics are well done and robust, as for the tables (Perhaps the format of the figures could be improved, but it is correct) I would change the format more in line with this journal. I have made small corrections and requested some doubts that have arisen when reading it

Annotated reviews are not available for download in order to protect the identity of reviewers who chose to remain anonymous.

---

## Round 0.2 · accepted · Accept

Two minor issues:

For the sentences: "The World Health Organisation has reported ground reflection of UVR from grass/soil/water is 10%, dry sand 15% and sea foam is 25% (33). Therefore ground reflection from water provides further rationale for the higher prevalence of MM in surfers."

This last sentence does not make sense. Would this not be better:

The World Health Organisation has reported ground reflection of UVR from grass/soil/water is 10%, dry sand 15% and sea foam is 25% (33), providing further rationale for the higher prevalence of MM in surfers.

For the sentence: "Our findings for skin cancer examinations indicates the majority (68% - 74%) of our participants underwent regular skin screenings whereas (41) De Castro-Maqueda ..."

The (41) should come after De Castro-Maqueda